# The Future of Climate-Resilient and Climate-Neutral City in the Temperate Climate Zone

**DOI:** 10.3390/ijerph19074365

**Published:** 2022-04-05

**Authors:** Patryk Antoszewski, Michał Krzyżaniak, Dariusz Świerk

**Affiliations:** Department of Landscape Architecture, Faculty of Agriculture, Horticulture, and Bioengineering, Poznań University of Life Sciences, Dąbrowskiego 159 Street, 60-594 Poznań, Poland; patryk.j.antoszewski@gmail.com (P.A.); michal.krzyzaniak@up.poznan.pl (M.K.)

**Keywords:** UHI mitigation strategy, UHI intensity, BGI, PET, built-up environment parameters, urban space parameterization, urbanized environment, street canyon, climate changes

## Abstract

The urban heat island (UHI) effect is the main problem regarding a city’s climate. It is the main adverse effect of urbanization and negatively affects human thermal comfort levels as defined by physiological equivalent temperature (PET) in the urban environment. Blue and green infrastructure (BGI) solutions may mitigate the UHI effect. First, however, it is necessary to understand the problem from the degrading side. The subject of this review is to identify the most essential geometrical, morphological, and topographical parameters of the urbanized environment (UE) and to understand the synergistic relationships between city and nature. A four-stage normative procedure was used, appropriate for systematic reviews of the UHI. First, one climate zone (temperate climate zone C) was limited to unify the design guidelines. As a result of delimitation, 313 scientific articles were obtained (546 rejected). Second, the canonical correlation analysis (CCA) was performed for the obtained data. Finally, our research showed the parameters of the UE facilities, which are necessary to mitigate the UHI effect. Those are building density and urban surface albedo for neighborhood cluster (NH), and distance from the city center, aspect ratio, ground surface albedo, and street orientation for street canyon (SC), as well as building height, material albedo, and building orientation for the building structure (BU). The developed guidelines can form the basis for microclimate design in a temperate climate. The data obtained from the statistical analysis will be used to create the blue-green infrastructure (BGI) dynamic modeling algorithm, which is the main focus of the future series of articles.

## 1. Introduction

### 1.1. The Urban Heat Island (UHI) Effect–Causes, Effects, and Countermeasure

The UHI effect describes the solar radiation balance, which for urbanized areas is low compared to non-urbanized areas. The UHI intensity is based on the temperature difference between the measurements taken in the city and at reference stations outside the city. Measurements are performed in various scales depending on the research needs [1,2] and using terrestrial equipment or remote sensing. Over a century of observing, the UHI effect has become the best-described climate change phenomenon [3,4,5,6,7,8,9,10,11,12,13,14,15,16,17,18,19].

For millennia, humanity has been modifying the natural landscape by transforming its coverage patterns for its own purposes. However, in recent decades, the increased intensity of these changes related to rapid and uncontrolled urbanization contributes massively to natural environment degradation and rises with city expansion [20,21,22]. Recently, the aggressive urbanization effects in the form of the UHI effect have increased and the emergence of local climate anomalies have been noticeable. For this reason, it is widely believed that increased city temperatures are a direct result of excessive urbanization.

A positive radiation balance is recorded in all cities throughout the year [23,24,25,26]. However, the UHI effect intensity shows diurnal, seasonal [4,6,27], and geographical variability, which will differ for cities in various climatic zones [28,29]. Solar radiation intensity, air temperature, relative humidity, and wind speed affect UHI appearance and its course [9,30,31,32,33,34,35,36]. The distribution of the UHI effect inside the city and its intensity depend on the degree of development of the area, interior geometry, and materials used [13,37,38,39,40,41]. Urban factors affect, among other things, increased solar exposure, absorption of short-wave radiation and trapping of long-wave radiation, reduction of the radiation reflection coefficient, increase of the radiation absorption rate, heat capacity, shift of the emission phase, changes in the dominant winds patterns, reduction of wind speed and its humidity, reduced evaporation, post-transport cooling, and turbulent and convective heat transport [42,43,44,45,46,47,48,49,50,51]. In addition, the built-up area density and the anthropogenic heat emission (buildings heating and cooling, transport and industry fumes emission, and general air pollution) [52,53] increase the latent heat gains. There is also a dangerous cause-and-effect loop in this case. The increased UHI intensity increases the temperatures inside the buildings and forces their occupants to cool them. On the other hand, the building’s cooling causes an additional increase in the temperature outside the building. The cause-and-effect loop closes, increasing the heat gain in the city canyons.

The UHI increase caused by the deteriorating quality of the urban environment impacts the overall increase in air temperature in the city, air circulation reduction, heatwaves intensity and frequency, and the concentration of the pollutants [54,55,56,57,58,59,60]. Those factors reduce the internal and external thermal comfort referred to as physiological equivalent temperature (PET) and increase morbidity and mortality [61,62,63,64,65,66,67]. In addition, they can be associated with health problems such as exhaustion, heat shock, thermoregulation processes disturbance, and cardiovascular stress [68,69].

Global warming will undoubtedly increase the frequency, duration, and intensity of related heat waves [70,71]. Therefore, it is necessary to counteract its formation now, bearing in mind that UHI mitigation also reduces the climate change rate [72,73].

### 1.2. Countermeasure Strategies

Researchers related to urban planning, architecture, and climatology are developing the city-UHI relationship’s mechanics and effective mitigation strategies based on the already-known dependencies [74,75].

Many tools have been developed to mitigate the elevated city temperature related to (1) reducing the emission of anthropogenic heat, (2) increasing albedo and evapotranspiration, (3) introducing nature-based solutions such as the blue and green infrastructure (BGI), (4) the use of regenerative design practices taking into account the local climate, and (5) modification of city geometry and de-urbanization [76,77,78,79,80,81,82,83,84,85,86,87,88,89,90]. However, the most efficient and striking root cause solution seems to be creating the BGI, which is introducing and increasing greenery and water object areas [91].

In previous studies, the researchers carried out a qualitative systematization of the BGI object parameters influencing the UHI effect mitigation (Table 1).

It has been proven that the BGI solutions, understood as vegetation and water objects, can significantly affect the urban microclimate by reducing the ambient temperature and increasing humidity. In addition, greenery definitely affects air quality by absorbing harmful gases, carbon dioxide, aromatic hydrocarbons, and dust, while replenishing the air with oxygen, essential oils, and alleles and ionizing them. Requiring supplementation is the development of the topic from the degrading side, i.e., examining the quality parameters of the urbanized space elements affecting the UHI intensity increase. Understanding the context and its impact on the local thermal image is necessary for the most accurate selection of the BGI parameters to reduce the ambient temperature.

This review article aims to systematize the geometric, structural, and spatial parameters of the elements of the urbanized environment, which may increase the intensity of the UHI effect in urbanized areas of the temperate climate zone. This review is a continuation of a series of articles describing the creation of a dynamic algorithm using the BGI solutions to control the city thermal landscape in the temperate climate zone.

## 2. Materials and Methods

Over the last 2 decades, many scientific papers have been published presenting the current scientific achievements related to the city elements’ influence on the PET and UHI effect (Table 2). Most articles, however, describe achievements in a reasonably narrow range, within a particular branch of parameters, a given scale, and different climate zones. Moreover, authors often focus on uncontrollable elements, forgetting possible architectural and urban design changes.

The urbanized environment parameters influencing the temperatures inside the city are highly interactive with the city’s climate. Climate describes annual geographical location weather patterns. As a result, it is an essential uncontrollable factor in urban planning [109,110] that must be followed. The lack of holistic systematics of design parameters for a given climate is necessary to fill. Such an approach will increase the normativism of research [111].

The delimitation of the UHI crisis area allowed us to designate the climatic zone marked by the most significant civilization imprint [112]. Since historic times, the temperate climate zone has been associated with a high degree of urbanization, and its cities have a high completion degree [113]. Those factors translate directly into the highest density of human settlement occurrence in that climate area and the highest intensity of UHI within its borders [114]. Therefore, it seems justified to select a temperate climate as a criterion for selecting studies for the review.

### 2.1. Selection Criteria for Papers

The bibliographic query was carried out to limit the number of scientific articles to those that most fully describe the impact of modulating individual parameters of elements of urban space on UHI. Therefore, only case studies from the temperate zone of the last 2 decades were included in the presented review paper.

For the selection of the scientific paper, the normative procedure for systematic reviews in the field of the UHI effect was used [115,116]: (1) a broad search criterion was identified to obtain many articles, (2) the number of research articles was limited to those eligible for the detailed criteria, (3) the required information was collected from the results of selected studies, and finally (4) a discussion of this research and its conclusions was presented.

The climatic delimitation was performed with the re-analyzed algorithm modeling the climatic distribution in the Köppen–Geiger classification [117]. In addition, the case study locations were analyzed if they belonged to the temperate climate in the year of publication. Therefore, the borderline climate cases have been included in the review.

The logical diagram of the bibliographic query is presented in Figure 1.

### 2.2. The Systematizing Elements Structure

The city should be understood as an ecosystem category because it is deceptively similar to a forest or a coral reef despite the significant differences in the building materials. However, for the ecosystem to function correctly, it must be in a state of dynamic equilibrium—a climax that cannot be noticed in modern cities. It took a lot of time to develop the complex relationships that led to the equilibrium state of ecosystems. Humanity slowly begins to understand the dependencies that it creates when building a city and what parameters of individual city elements influence one another. This barely discovered network of connections creating deep synergies between elements and parameters [118,119,120,121,122,123,124,125,126,127] shows how much solid work scientists and designers must do, understand, and learn to translate the laws governing the city’s climate for the purpose of design.

To better understand the spatial and microclimatic relationships occurring in an urban unit, it is worth applying the islands’ theory in terms of landscape. For this reason, this chapter’s sections are logically divided into four scales corresponding with three different landscape formations with a significant impact on the microclimate. The scales are hierarchical, and each item lower in the hierarchy is a component of a higher scale (Figure 2). The smallest scale is related to the properties of materials, i.e., the basic physical building blocks. It is related to morphological parameters and describes the quality through the properties of materials that make up elements and objects at various scales. It is problematic to classify the parameters responsible for anthropogenic emission. They define the de-facto quality of the space and have also been classified into the family of morphological parameters. Higher in the hierarchy, a building corresponds logically to the tree. The third scale, or city canyon, is made of the buildings and the street layout, just as trees and the river make up the valley. The fourth and last scale corresponds to the neighborhood unit, where buildings, streets, and other elements of the urban environment create complex spatial patterns, as well as biotic and abiotic components in a forest massif. These three scales represent a variety of geometric family parameters. There are significant differences in the types of parameters, as well as in the different distribution of the parameters’ dominance in the family. The geometrical parameters family describes quantitative parameters (mathematically countable) synonymous with length, width, height, area, volume, distribution density, percentage, area, and interrelationships. Extra scale, there are also universal temperature modulators, identical for the elements and responsible for the energy balance, determining the function of space, the geometry of the entire unit, and the external factors that influence the objects on the scale. It is the most problematic scale closely related to the group of topographic parameters. Its problems result from the inability to assign to the scale because, depending on the hierarchical level of the scale, individual geometric parameters become topographic. In this way, on the scale of the estate, all geometric parameters for the entire city become topographic parameters for the estate because they become external factors—the context.

It may be controversial to include the turnover parameter in the family of topographic parameters, despite the possibility of assigning urban units to scales. However, when looking at the nature of the influence of this parameter on solar gain, it can be seen that the influence of this parameter is fully dependent on the temperature depending on the position of the sun on the horizon. This relation brings to mind other factors classified as external to the scale, like the distance from the sea and the associated gain in air humidity.

### 2.3. Statistical Analysis of the Results

A zero-one distribution was used to prepare the matrix of results and interpolate the data from the literature review into numerical data. Each publication was treated as one n, and then a matrix was constructed containing repetitive variables for:(a)Elements of the built environment: neighborhood cluster, street canyon, and building;(b)Parameter family: geometric, morphological, and topographic parameters;(c)Country where the research was conducted.

The above-described method allowed an advanced statistical technique for this type of descriptive data.

Statistical analyses were based on the discriminant analysis. The result of the analysis was to check which of the families of parameters are the most important for the UE and countries most often tested.

Ordering techniques were used, ordering the test samples along the gradient represented by the ordinate and the abscissa. A compliance analysis (CA) was performed to check which technique would be most appropriate for the analyzed dataset. This procedure was to respond to the nature of the structure of the analyzed dataset based on the gradient length (linear or unimodal). The gradient length (>3) suggests that the canonical correlation analysis (CCA) is suitable for this dataset type.

The families of parameters and their testing frequency were compared in discriminant analysis. It was also checked which parameter families are most relevant for the UE. For this purpose, a progressive stepwise analysis was used. All variables were assessed. The variables that contributed most to group discrimination (based on *p* and F values for each analyzed variable) were included in the model. This process was repeated until the *p*-value dropped below 0.05 for the studied variable. The Monte Carlo permutation test was performed to determine the significance level (separately for each variable and then for the entire model). All tests, calculations, and graphic elements were prepared in Canoco for Windows software and Microsoft Excel spreadsheet. The following tools were used with Canoco for Windows: Canoco for Windows 4.5, CanoDraw for Windows, and WCanoIMP.

A Pareto diagram was built to determine which parameter families appeared most often in scientific papers.

## 3. Results

The results of the research are presented in Table A1 in Appendix A. The specified groups of parameters, geometrical, morphological (intra-structural), and topographical (depending on the context) for individual elements of the UE, contain individual parameters and their occurrence in the literature.

### Statistical Analysis Results

The significance of individual parameters was expressed as a percentage, depending on the number of articles. The analysis results (Table 3) showed that the research on the UE influence on the UHI propagation in a temperate climate mainly focused on geometrical parameters belonging to two elements of the UE: neighborhood cluster and street canyon (Figure 3). The fewest citations concerned the group of topographic parameters, which can be explained by the relatively small range of this group of parameters. In the literature, the most frequently quoted geometry parameters are the basic indicators describing the elements of the UE.

For the neighborhood cluster (NH), the most crucial parameters are for the geometry parameter family–building density (BD), the morphology–urban surface albedo (WAS), and the topography–the distance from the city center (CBD).

The following were highly important for the street canyon (SC) structure for the geometry–aspect ratio, morphology–ground surface albedo, and topography–street orientation.

Finally, the most critical parameters for the building structure (BU) were in the group of parameters: geometrical–building height (BH), morphological–material albedo, and topographical–building orientation (O).

The CCA analysis showed that the most frequently described UE objects were the neighborhood cluster (NH) and street canyon (SC), as indicated by the exceptionally high F-values (Table 4). Both for the NH and SC, geometric parameters turned out to be significant, while for the BU, morphological parameters turned out to be substantial. Most often, information on the UE elements influencing UHI propagation can be found in the papers with examples from China, Italy, the USA, Greece, Germany, and France. The distribution of publications depending on the country of origin is very scattered. However, some dependencies can be found. The most frequently described geometric and topographic parameters for the street canyons were in Brazilian, Portuguese, Syrian, Algerian, and Tunisian research papers. In Indian, Serbian, Danish, Vietnamese, Polish, and Czech papers, elements related to the geometry of the neighborhood cluster were studied more often. Buildings, i.e., the smallest units, were the object of research in such countries as Italy, France, Switzerland, and Lebanon (Figure 4).

Although we have determined the most critical parameters of the urbanized environment elements, the modulation of which can bring the most significant changes in the UHI effect, many more design variables are involved in creating the thermal image of the city. The section below describes all the essential modulators, synergistic relationships they create, and how they affect the intra-city PET and UHI.

## 4. Discussion and Urban Design Strategy Recommendations

### 4.1. Out-of-Scale Parameters

The basic property of urban space is its solar exposure influencing the internal energy balance [128]. This is referred to as the sky view factor (SVF). By scientists in general, SVF was considered the basic indicator determining mainly the geometry of the city canyon. However, SVF is a qualitative parameter; therefore, its translation into urban planning is problematic [129]. To understand what is responsible for the city’s chiaroscuro distribution, we need to break SVF into smaller elements. Land cover with buildings [130], building density [130,131], building volume [130], aspect ratio [132,133], building symmetry [131], building element proportions [130], building orientation [134], building height [134], and the presence and size of trees [135] and other shading elements [136] are parameters that build up SFV. SVF is positively associated with land surface temperature (LST) [137,138], atmospheric temperature (AT) [139,140,141,142,143,144,145], and the UHI effect [146,147,148,149], both during the day [150,151,152,153,154,155,156] and at night [157,158,159,160,161,162,163,164] in all seasons [165]. Only after a deeper analysis, can the complicated pattern of dependencies governing SVF be seen. The availability of the sun changes with the season changes, and the growing season seems to have an obvious influence on this change [159]. The same level of SVF affects incoming radiation during the day and outgoing radiation at night [159,164]. At the same time, it can hinder sunlight access and retain the heat accumulated in the urban tissue. Moreover, solar availability is a basic element influencing the albedo efficiency [166], emissivity [167], and convective air movements related mainly to the heating of various surfaces [168,169].

The type of land cover is partly responsible for the temperature distribution within the city. Different city zones affect PET and UHI differently because of their function and class [170]. The type of coverage is characterized by its specific anthropogenic emission, pollutant concentration, a combination of spatial parameters such as the density of buildings and roads, and the size of impervious surfaces that receive solar radiation. For this reason, residential, commercial, and industrial zones have a particular impact on increasing intra-city temperatures [171,172,173,174]. The industrial areas are characterized by the highest pavement temperatures, high gas emissions, and energy consumption, increasing the daily anthropogenic heat. Accordingly, they are considered the main UHI propagation points [175] with a limited contribution to the overall city’s UHI effect because of the small total area [171]. Commercial spaces are also characterized by high anthropogenic heat. In this case, it is result of high daily energy consumption, traffic, and crowds of people [176]. Finally, the residential area may be the largest radiator of anthropogenic heat during the day, mainly from space heating and cooling [177]. This zone, similar to the office, education, health, tourist, and religious zones, usually has a low built-up area density and high green area coverage, which reduces the impact of heat emission on the UHI effect [170]. The most favorable type of cover for PET and UHI are green areas, agricultural lands, and water objects [178,179,180,181,182,183].

The thermal behavior of an urban unit is directly influenced by its size. In the forest case, its size is of great importance for the inside temperature and humidity amplitude. However, this dependence is inversely analogous for urbanized areas. Research shows that the larger the city, the higher the temperature. However, one must deal with a specific hierarchy resulting from the original city size in this case. Increasing the large city size will bring more significant changes than increasing a small city size. It happens because of the number and density of urbanized areas showing higher temperatures and the synergy among the elements [118,137,172]. The densification of various urbanized areas may disturb that relationship’s linearity. Aggregated smaller cities may have a higher UHI intensity than large cities with low building density [184].

Another temperature modulator of an urban unit is its degree of centrality. A greater degree of centralization shows many city hotspots and thus the higher UHI levels. This dependence is built by spatially complex street layouts with a high degree of irregularities, with more intersections, greater density of impervious surfaces, a more significant number and density of buildings with irregular distribution and shapes, and the related anthropogenic heat [185,186,187] characteristic for a centralized city. This description perfectly reflects the center’s image of an old and large city [136].

The distance from heat sinks, both cold and hot, affects the temperatures inside an urban unit [151]. The proximity to the unit’s center (a city or a housing estate) increases the local temperature [188,189]. Distance to a cooler green area heat sink is impactful throughout the day [154]. The closer the green areas are, the cooler it is during the day [190] and it is slightly warmer at night [188]. A similar relationship can be noticed concerning water objects [191]. These diurnal differences are related to the thermal capacity of water, heat retention under the canopy of trees, and the landscape context [192].

### 4.2. Material Properties

This scale shows the thermal behavior of the city’s basic building material and the impact of its thermal properties on the immediate and farther surroundings. In this context, the significant thermal parameters of building materials include surface permeability for water, heat capacity, albedo, insulation and conductivity degree, absorption rate, diffusivity, and emissivity.

One of the essential material parameters is the permeability or porosity of the pavement. This affects the intensity of evapotranspiration and has a significant impact on the temperature of pedestrian and road surfaces [193,194], the UHI effect [195], and the high AT duration [176]. Due to their structure, green areas [142], both with predominantly high or low greenery, are characterized by the highest degree of transpiration [155,193]. The effect of evaporative cooling from permeable surfaces is noticeable within 150 m. Since the intensity of evapotranspiration is influenced by the temperature of the city air, the lower temperature can be felt, especially in summer. Literature reports that the daytime temperature may drop by 3.4 °C and night temperature by 1.2 °C because of the use of permeable surfaces [195].

Besides its evident influence on building temperature, thermal insulation impacts the city air temperature because of the energy exchange between the building and the canyon [196]. Poor thermal insulation increases the intensity of this exchange. The impact of this parameter on the city’s energy balance becomes more important, especially in winter, when building heating prevails [163]. The size of the heat exchanger is not insignificant. With the increase of the building wall area determined by the canyon height, the effect is multiplied, which is noticeable in the rise of the canyon temperature [163,197].

The decrease in the thermal conductivity of the pavement reduces heat absorption [198] and contributes to the formation of higher pavement temperatures [199]. In turn, increasing the conductivity parameter contributes to greater heat transmission into the pavement, which may be dangerous for underground infrastructure [193,200].

Increasing thermal diffusivity reduces the daytime ambient temperature and raises the night temperature [147]. An example of a material with high diffusivity is a granite surface [193].

The radiation absorption rate is another parameter that strongly influences the pavement temperature. Reducing the absorption rate can significantly lower the temperature of the material [199,201] and bring more significant thermal benefits to the city than the increase in emissivity [199].

The materials’ heat capacity depends on, among other things, the density, mass, and material albedo [202]. For this reason, both hard construction materials such as asphalt, concrete, and dense bricks and natural surfaces such as grass/soil and water are characterized by high heat capacity. These surfaces have a greater tendency to store large amounts of heat in their volume. This property affects the time shift in heat release because the heat absorbed during the day will be released to the atmosphere at night [203,204,205]. The capacity of the pavement can be increased by manipulating the density and thickness of the pavement foundation. Increasing these parameters causes an increase in thermal capacity [206]. On the other hand, its decrease is caused by an increase in the material’s porosity, i.e., its density, in fact [206]. Thermal mass, which has similar dependencies, has a more significant impact on the city-scale temperature [44]. In this case, it is possible to manipulate the building’s density, volume, and area. A higher building density similarly increases the area thermal mass [155]. On a larger scale, the stored heat does not affect the average air temperature but significantly affects the amplitude and phases [44]. The stored heat reduces the city temperature fluctuations but, on the other hand, it increases the intensity of UHI [163,200,201]. Moreover, the district’s high heat capacity may increase anthropogenic heat resulting from cooling rooms in the summer [184]. For example, old city centers tend to have a high thermal mass, and therefore PET in these districts during the day is much lower, and nighttime UHI is much higher than in newer districts [155,207]. The thermal mass can be modulated with the canyon’s shape [48] by increasing or reducing the area of the walls of the building [155]. However, this dependence is only available for canyons with a high degree of surface development [208].

The urban surface emissivity results from the physical and chemical material structure [209]. It has a significant impact on the energy balance of the canyon, influencing the total radiation emitted from it [196,209]. During the day, the emissivity influence is invisible. At night, it can cause temperature drops [147,199] even by 0.3 °C with a change of 0.1 in emissivity unit [124]. The emissivity can be modulated by the housing estate geometry [167], using so-called cold materials [210], or painting the material surface with special film-forming paints [211]. Decreasing the emissivity of roofing materials may reduce a building’s thermal losses [211]. Thus, reducing emissivity may decrease anthropogenic heat emissions caused by heating or cooling buildings.

#### Material Albedo

The albedo is the ratio of reflected radiation to incoming shortwave radiation [212]. In other words, it is a material’s ability to reflect solar radiation from the city surface [213]. Therefore, the influence of albedo can be considered only when the surface is exposed to high sunlight during the day [166]. Other essential elements, such as greenery, take control of the temperature in the case of sunlight absence [161,183]. The higher albedo leads to lower surface and air temperatures [132,161,214] and a shorter duration of high temperatures. Similarly, materials with lower albedo, e.g., asphalt, increase the ambient temperature and hinder cooling [176]. Research shows that increasing the global urban albedo by 0.13 units can lead to a temperature drop of up to 4 °C [196]. The urbanized area differs in albedo distribution. Its distribution in the unit defines LST and PET [215,216]. The albedo increase is possible because of a decrease in the density of the buildings [217], structure irregularities of housing estates [218], the buildings’ heights, and an increase in building height uniformity [217] and the bituminous aggregates exposition [219]. The albedo decline is associated with age and material consumption [219]. By varying the albedo level in an urban unit, it is possible to improve human thermal comfort [188]. Albedo can soothe both day and night UHI effects [151,220]. The world of science is divided here. In some studies, albedo has a more significant impact on daytime temperatures [151,161,221]. Other studies indicate a more substantial albedo impact at night [188,212]. A settlement of this dispute may be the fact that there is a critical point in the afternoon. At this point, high albedo materials, by reducing sunlight, lose their influence in favor of low albedo materials that initiate radiation re-emission into canyons [188].

Albedo is highly correlated with building density. Low proportions of canyons create better conditions for a higher albedo [213]. In addition, the use of high albedo materials in scattered housing and the low aspect ratio (H/W) canyons have more significant benefits [132,222] because of prolonged ground surface exposure to solar radiation. In the case of the same albedo level, the temperature will be lower in the square than in the courtyard [133]. The use of a high albedo surface in a dispersed context may improve thermal comfort at the pedestrian level [222], and alleviate UHI [212,223], but also lower the ambient peak temperature [183,224] and thus also the cooling load in buildings [196,224]. Research in Australia has shown that for every 10% albedo increase, the decrease in peak cooling load was about 3.5% [224]. The USA example shows that increasing the parking lot albedo can lower temperatures by about 1 °C [223]. Increasing the albedo in densely built-up areas is unwise because it implies an increase in the mean radiation temperature [214,225]. This is due to radiation reflection multiplication in the canyon [226]. Increasing the ground surface albedo increases the temperature of the walls [225,227]. The use of high albedo on the walls lowers the walls’ temperatures and raises the road temperature [205]. In this case, the wall temperature increase may equal 3.5 °C [227]. This situation may exacerbate thermal and optical discomfort and increase cooling energy consumption by reflecting the radiation inside the buildings in the canyon [166,227]. Thus, it may directly affect the UHI intensity increase [161]. The solution to this problem is either lowering the street albedo or using retroreflective (RR) materials in the canyon [228]. This coated material has a high equivalent albedo in any urban context and does not multiply the canyon radiation. When the light falls on the RR material, it is reflected in the same direction. In this way, it is possible to avoid the radiation scattering effect in the city canyon specific for high albedo materials and reduce the absorption of radiation in the urban areas at a level that balances the anthropogenic emission [218].

The increase in the roofs’ albedo does not reduce ground-level air temperature [197], but it significantly reduces the LST [229,230,231]. Using a phase change material (PCM) roof coating may lower its temperature by 20 °C [173]. This is also an effective method of reducing peak cooling and heating loads in buildings [229,230,232], especially if the entire building is covered with high albedo materials [197]. The intensity of the UHI effect mitigation caused by the roofs’ high albedo can be forced by increasing the roofs’ area and building density, and reducing the heights of buildings [173,231,233].

### 4.3. Building Scale Design Parameters

This scale shows the influence of building parameters on both its internal and external thermal behavior. On this scale, one can notice the significant impact of two groups of parameters on temperature changes. Those groups are the building geometry determined by its surface, shape, height, and anthropogenic emissions related to air-conditioning use and energy efficiency.

#### 4.3.1. Envelope Geometry

The building area size significantly affects the UHI effect [234,235,236], both during the day and at night [153]. Increasing the building area leads to an increase in LST [237] and extends the duration of high air temperature. This relationship is more evident in commercial than residential buildings [176]. Despite the negative impact of buildings’ large area on temperature, in the initial phase of increasing the building floor area factor, a reduction in the heating energy consumption can be noted [238]. This means that the ideal proportions of the building area should be found, which will not significantly affect the LST and reduce anthropogenic emissions.

The shape of a building influences the local thermal environment and daytime comfort in different ways [131,174]. An increase in the complexity of a building shape may lead to the LST increase because of increased solar exposure of facades [190]. However, the increased facade surface area may contribute to more intense energy exchange between built-up areas and vegetation [190,239]. In addition, shortening the length of the building itself may improve thermal comfort and help avoid the occurrence of ventilation shade on the other side of the building [54]. The above dependencies suggest that buildings with compact and regular plans are the best for the external thermal environment if there is no greenery near the building [240].

Building height is one of the most critical factors affecting temperature [123,178] and UHI [170]. It influences the local thermal environment around the building, changing the conditions of insolation and ventilation [241,242]. As the building height increases, the LST and the daily temperature amplitude increase [164]. Buildings with a height of 10 m show the highest temperature amplitude during the day [123,243]. The significance of the building height is strongly dependent on solar exposure [122,126], which is modulated by the density of the surroundings [129,244]. For this reason, the temperature amplitude of the high parts of the building will always be higher [164]. In the case of a pillar-based building, raising the height of the building increases the wind flow underneath the building, especially in the corner zones, which slightly improves the PET around the area [245]. Increasing the height of a highly exposed building increases the evapotranspiration cooling effect of nearby trees [54,178]. The building height by shadowing determines the energy consumption of buildings [246].

#### 4.3.2. Anthropogenic Emission

Japanese researchers believe about 40% of anthropogenic heat comes from buildings [247]. This contributes to UHI intensity and the thermal comfort deterioration at the pedestrian level, and extends the duration of the elevated temperature [135]. The heat emitted from buildings depends on their energy efficiency. A building’s energy efficiency is influenced mainly by the percentage of the building envelope, its material, and daytime sun exposure [246,248]. It depends less on the patterns of wind flow and air temperature [48,248]; therefore, the context contribution of the building location cannot be ruled out in this relationship. A building in a densely built-up area will be more energy efficient than an insulated building [249], especially if there are large roads in its immediate vicinity [227]. Excessive anthropogenic heat emitted from a building is highly correlated with two seasons: summer and winter. In winter, space and water heating are responsible for emissions [197]. In summer, the air conditioning used to cool rooms takes over [177,184].

Air conditioners installed outside the building affect local heat fluxes. They can raise the building’s ambient temperature by up to 1.7 °C [184]. Their effect on temperatures appears to be paradoxical and self-perpetuating. The increased temperature outside necessitates cooling inside the building. In this case, the heat emitted from the A/C heats the air outside, increasing the need to cool the building. In this way, air conditioners can significantly increase heat wave mortality [177]. Studies in France and Hong Kong show that this unfortunate loop can be broken by managing air conditioning in rooms [135,177]. Energy-saving A/C operation helps to reduce the emission of anthropogenic heat, the intensity of UHI, and the duration of elevated temperatures by up to 28%, especially at night [135].

Using a green roof or a cool roof [224] will help reduce both the cooling and heating load, the latter being more effective [231]. The green roof also has a slight influence on the ambient building temperature. However, effective space cooling depends on increased context density and its albedo decrease [183]. The problem of this solution is the maintenance itself in climate varieties that are less favorable for vegetation [250].

### 4.4. Street Canyon Scale Design Parameters

The scale of the city canyon includes buildings and a road, square, or other element forming its bottom. It may consist of both a canyon in the sense of a street with a parallel arrangement of buildings and closed courtyard arrangements. Elements in this scale have both local and supra-local impacts. The obvious factor to discuss is the canyon’s shape, which affects all its thermal relationships. In addition, the influence of orientation and shading elements on the local thermal environment of the canyon is noticeable compared to others.

#### 4.4.1. Surface Geometry

The mutual height ratio to the canyon width is referred to as aspect ratio and denoted as H/W. This parameter is responsible for the cross-sectional geometry image of the street, square, or courtyard and tells about its depth/openness and, in a way, solar exposure. The canyon height and width modifies the thermal conditions in the street space [201,251,252,253]. One can observe a special relationship between the elements which create the canyon aspect ratio. The greater the canyon height, the narrower the canyon, and the higher the H/W ratio, the greater the canyon depth. The H/W influences the shading patterns [170] and wind [47,133,170], thermal mass [155], and canyon radiation balance [155]. It also significantly influences the LST, canyon air temperature fluctuations [133,137,155,163], the occurrence of the thermal peak at the pedestrians level, their thermal comfort [47,254], and less significantly the UHI [142,255]. It should be noted that the influence of the shape factor is more dominant in the case of road surface temperature than in the case of the roof temperature [152]. However, equalizing the canyon height and width blur this difference [164,207].

The canyon geometry dynamically influences its thermal behavior during the day and throughout the year [227,256,257,258,259,260], and changes in canyon depth have better mitigation potential for the nocturnal UHI [121,188]. Wide canyons are characterized by a rapid LST increase in the morning, and because of the greater heated surface area, they reach higher maximum temperatures earlier [227,261,262]. At night, their thermal response is better than the high canyons. The low H/W ratios canyons have much greater net longwave radiation exiting the canyon and more significant convective cooling [257,261]. As a result, they cool down faster and show a greater proportion of cool area than high H/W ratio street canyons [201]. That explains why wider canyons have a smaller overall UHI than deep canyons [207,262] despite a greater intensity variation during daytime [261,263]. As the aspect ratio increases in the morning, the UHI effect becomes the urban cool island (UCI) effect because of the strong shading effect [132,135]. By reducing the penetration of direct sunlight [227], higher H/W allows the bottom of the canyon to remain completely shaded and thus to heat slower [256]. The maximum short-wave radiation in such a canyon occurs around noon [227]. The increase in shaded areas results in lower maximum temperatures [155], daily temperature ranges [201], the mean radiant temperatures (T*_mrt_*) [133], and the LST [142,146,264,265,266], especially walls with greater sun exposure [261]. Human thermal comfort is also improved [133,222]. That can be seen in the courtyards case. Their deepening can lower their air temperatures by about 2–2.5 °C [267,268]. The deep canyon is cooler in the afternoon than the shallow one [269]. However, the situation changes in the late afternoon. At night, the higher street aspect ratio and symmetry result in less outgoing net longwave radiation by reducing SVF, air cooling, and multiple radiation reflections inside the canyon, resulting in a thermal trap effect [242,248,256]. Those increase the air temperature [265,269,270], PET [254] and the UHI [227,263,270] compared to shallow and open canyons [127]. Even a small change in the canyon geometry at low shape factor values can cause heat island condensation [155]. However, the H/W impact on the UHI only gains importance for buildings higher than two stories [181] or a ratio greater than 1.3–1.5 [132,239]. The canyon smoothness and symmetry also contribute to a slight temperature increase at night [131]. The seasonal relationship correlates with solar access variability and is more pronounced in summer than in winter, depending on the different angles of sunlight incidence [256,258,259,260,271,272]. The temperature difference between the deep and shallow canyons drops significantly during the winter [256]. Due to the greater sunlight access, shallow canyons in winter have the highest daily T*_mrt_* and are more comfortable than deep ones [119,256]. However, the increased street width does not improve thermal comfort equally in all orientations during the winter [253]. Increasing the building height and reducing the street width initially increases the minimum and average daytime and night temperatures [273]. The lower solar angle and the higher H/W ratio of the courtyard increase the reflected long-wave buildings’ radiation and reduce heat dissipation. This effect can increase with the building’s height until the H/W ratio is 5.5 [258]. Above this ratio, less and less sun reaches the canyon bottom.

The relationship between the canyon geometry and the wind environment is quite complex. It affects both the wind speed [133] and the nature of the canyon air circulation [47,170] and can cause mechanical turbulence [124]. The importance of geometry becomes apparent when one intends to use wind cooling [270]. In open canyons, the convection cooling quality is more significant than in narrow and closed canyons [133]. However, this relationship can be modulated. When the wind directions are parallel to the canyon, it is possible to increase the wind speed by increasing the H/W ratio [133,274]. In this scenario even a deeper street canyon allows weaker wind loads to penetrate to the pedestrian level [275]. Where the predominant wind directions are transverse to the canyon, great depth is not suitable for pedestrian level ventilation [257,275]. The wind speed reduction in this configuration is caused by the formation of vortices between buildings [227] and the increased air displacement above the roofs [256]. In this case, the high canyon smoothness and symmetry work unfavorably [131]. Increasing the asymmetry may be helpful [245]. The step-down canyons always have a higher temperature under high and low wind speeds than step-up canyons [276]. In addition, reducing the buildings’ heights at intersections to two floors may increase the canyons’ wind speeds. At the same time, this may reduce temperatures by 0.2 °C [250]. Care should be taken in modulating the wind environment to keep the canyon temperature down. In hot conditions of warmer climate variety, the H/W reduction can lead to canyon convection heating and warm air flowing from outside the urban unit [133,270].

The canyon geometry is not affected only by the aspect ratio itself [240]. The canyon space also has important thermal geometric features, mainly its symmetry. There can be various openings, recesses, protrusions, and the building’s facade elements, e.g., balconies or arcades [131,133]. Opinions on the effect of spacing between buildings along the street are divided. Some studies confirm their significant impact on temperature [271], others are negligible [190]. This is probably related to the prevailing wind patterns and street orientation. In studies conducted in China, wider spacing between buildings along the street resulted in poorer thermal comfort for pedestrians [264]. In studies conducted in the USA, the ground surface temperature decreased with increasing mean distances between buildings [190]. It is possible that several canyon configuration parameters may dominate the effect of spacing on the temperature. The canyon’s length does not significantly affect the thermal comfort at the level of pedestrians [47], but its elongation results in a more evenly distributed temperature during the day [277]. The effect of urban canyon width on the UHI effect may be multiplied by the simultaneous increase in the percentage of road coverage [154,170]. Wider streets also result in more significant traffic, which may intensify the UHI effect caused by increased heat and exhaust emissions in the canyon [170].

#### 4.4.2. Street Orientation

The street orientation affects the duration and intensity of solar radiation introduced into city canyons [155,202]. Therefore, its importance in modulating the canyon temperature is mainly influenced by the radiation intensity [222]. When a day is sunny, the orientation determines the air temperature and soil temperature. The orientation effect is neglected when the day is cloudy [120,278]. That relationship also occurs with regard to the seasons [132]. Canyon orientation is a parameter that produces intra-city thermal anomalies [155,226]. It affects thermal comfort [186,208,264], including size and duration of pedestrian-level thermal peaks [254]. Moreover, it shows a strong linear correlation with the temperature of the ground surface [208], walls [261], and the intensity of AT or UHI [186]. Street orientation determines the average ground radiation intensity stronger than the average facade radiation intensity [278]. Depending on the direction of the prevailing winds, the canyon’s orientation may also increase or decrease its speed [202] and increase the humidity [48], which affects the way the temperature is percepted.

By ingerention in the shadow areas, the canyon’s area density differentiates the daily air temperature between streets oriented in different configurations [269]. The orientation influence weakens when a canyon’s built-up area density and the H/W ratio decrease [208]. In a narrow canyon, orientation is not correlated or is slightly correlated with canyon air temperature, both in summer and winter [132]. However, it contributes to lower wind speeds [235], which positively concerns PET in the winter [132]. In addition, increasing the direction variation in the narrow street may result in less radiation infiltration into the canyon, irrespective of the street main direction [256]. In a wide canyon, orientation entails significant variations in wall surface temperatures during all seasons [132].

The N–S direction provides the shortest period of solar radiation in canyons [205], and the shading percentages on horizontal surfaces vary very little over the year. Hence, shading in the streets along the N–S axis is much more favorable than on the streets along the E–W axis, both in summer and winter [272,279]. Although the mean T*_mrt_* is similar in the N–S and E–W canyons [119], the number of T*_mrt_* hours exceeds the threshold values for moderate and robust heat stress. Therefore, the stress may be higher in the N–S canyons than in the E–W canyons [119,132,279]. In the N–S canyon, the opposite facades are equally shaded throughout the year, but the hourly temperature distribution of these walls varies. During the day, the western wall temperature rises faster. Still, the eastern wall, which has a higher solar load, shows greater amplitudes of daily temperatures and higher maximum temperatures [261]. The daily temperature amplitude of the east and west walls can be greater than that of the canyon air [261]. This relationship is visible above the fourth floor and on the ground floor (radiation from the ground) [280]. The lowest T*_mrt_* can be observed in the shaded areas of the east-facing walls [281]. The temperature of these walls rises until 10 a.m. and then drops [48]. The subsequent increase in T*_mrt_* starts on the west-facing wall around 3 p.m., but there is no heat stress until 5 p.m. From 8 p.m., there is a T*_mrt_* decrease [282]. The spatial N–S canyon air distribution and the WE and E walls become much more uniform at nighttime [261]. The best thermal comfort conditions for streets along the N–S axis are found for medium and high H/W ratio values (0.8–3.0) [272,279,283]. This way, it reduces the daily temperature amplitude and equalizes the differences in the diurnal temperature range (DTR) and maximum temperature of canyon walls [261]. For this reason, both the W and E canyon facades have more favorable conditions than the southern facades [279]. Increasing the H/W ratio significantly reduces cooling loads and increases heating loads in the canyon [284]. Increasing the spacing between buildings along the N–S street may lower T*_mrt_* and improve pedestrians’ thermal comfort [264]. Moreover, the shortening of N–S streets contributes to arcaded streets cooling [283].

The E–W direction provides the most extended duration of direct sunlight in the canyon and courtyard [272,279], and it is considered a warmer thermal configuration [269,279]. Due to the northern exposure of the canyon wall, the E–W orientation is exposed to sunlight from early morning. The result is the earliest maximum surface temperature [257]. Depending on the season, high T*_mrt_* only occurs for few hours, from 11 a.m. until approximately 3–5 p.m. [282], when it reaches its highest level. After this time, due to the solar radiation blocking, the absorbed heat is relatively quickly released into the surrounding environment [48,282]. However, it increases AT [257,272]; hence, thermal stress may be felt almost until 8 p.m. [282]. The increased H/W ratio for the E–W streets has no significant effect on the shading percentages [269,279]; therefore, it does not lower PET levels [222] and has little effect on daytime air temperatures [269]. On the E–W axis, the low H/W ratios (0.6–3.0) can therefore represent PET outside the comfort zone for most of the day [222]. With the H/W ratio above 3.0, it is possible to achieve a satisfactory daily thermal comfort level on the E–W streets [119,222]. A low H/W ratio (<0.6) is the most favorable in terms of solar gain in summer and winter for southern facades. As a result of short-wave and long-wave reflections, radiation emitted from the south-facing canyon walls can have an exceptionally high T*_mrt_* [281], especially on low floors [257]. The T*_mrt_* difference between the north and south walls can be 20 °C [119]. Increasing the H/W ratio on E–W streets reduces cooling loads and significantly increases heating loads [279,284]. Increasing the spacing between buildings along the E–W-oriented street increases cooling loads [284], and narrowing the spacing reduces T*_mrt_* and improves PET [264]. Increasing the canyon’s length in the E–W direction extends the duration of solar radiation. It increases the street temperature [250,285], but in the case of arcaded streets, it improves their cooling [283].

The shading intensity of diagonal streets is similar in all their rotation configurations. It is between the N–S and E–W axis streets’ shading intensity [279]. SE–NW street canyons provide more shade, higher wind speed [264], and better pedestrian thermal comfort conditions than other scenarios [264]. The H/W ratio impact on the number of sunlight hours suddenly decreases for street orientation angles of less than 30°. In comparison, when the street orientation angle exceeds 60°, the H/W ratio impact increases rapidly. The H/W ratio increase in the NW–SE canyons considerably reduces their daily temperature amplitude [269]. Diagonal streets with a H/W ratio between 1.5 and 3.0 can provide satisfactory thermal comfort conditions for most of the day [285], similar to streets on the N–S axis [222,279]. For H/W ratios higher than 1.3, diagonal canyons are similar to the E–W axis streets in terms of solar access to building facades in winter [279]. A closer look at the facade’s thermal dependencies in diagonal configurations shows that the maximum temperatures for the NE facade appear in the early morning. They are lower than the SW facades, where the maximum temperatures appear around noon. The NE facade minimum temperatures in the late afternoon are higher than SW facades, where the minimum temperature is measured in the early morning [196]. The SSW facade has higher maximum temperatures than the NNE elements. The most significant differences observed between these directions are visible in the afternoon and the lowest in the early morning [280].

#### 4.4.3. Canopy Properties

The trees and free-standing building elements’ presence in the canyon modifies its shadow patterns. During the day, this directly leads to a reduction in LST and PET [242,254]. At night, it increases its temperature by obstructing the long-wave radiation exit from the canyon [207,286]. This type additionally intensively reduces the wind speed at the pedestrian level [262,286]. The temperature level changes in street canyons associated with the trees’ presence depend on the tree crown-cover size [287,288], their planting density [245], their height [135], and solar exposure [267]. Small greenery elements in the canyon will only affect the local thermal conditions [285]. Due to more effective radiation shading and more substantial turbulent transport, tall trees have a more beneficial effect on the canyon temperature than small trees [135].

The city canyons’ geometry significantly modifies the trees’ thermal behavior in alleviating the street microclimate [286,287]. Due to changes in sun exposure, canyon shallowing and widening increase the trees’ cooling effect in the canyon. [264,287]. Research shows that high-tree-coverage streets can achieve better cooling quality when the H/W aspect ratio exceeds 0.67 [283]. For H/W aspect ratio = 1.0, increasing tree cover or reducing the spacing between trees may result in better tree shading and cooling during the day and a night temperature increase [285]. For H/W aspect ratio = 1.2 small and low-crown trees can generate a lower wind speed at the pedestrian level [286]. When the H/W aspect ratio = 2.0, both small-crowned and large-crowned trees may have the most significant air temperature cooling ability in daytime [271,286] and heating ability at night in the canyon [286]. When the H/W aspect ratio ≥ 3.0, the shadow cast by building walls can dominate the trees’ shading. Their influence on temperature reduction becomes less critical [286]. Dredging a canyon without increasing the number of trees reduces the cooling effect because the evapotranspiration cooling energy has to handle a larger air volume [208,267]. Adding trees in this situation could lower the temperature by as much as 4.5 °C on a summer afternoon [267,271]. For the high H/W aspect ratio, the diffusive planting of small-crowned trees favors street ventilation. It is a more preferable solution for summer cooling in these canyons than the higher density of trees and large-crowned trees [286]. This relationship is inversely proportional to the low H/W aspect ratio [135]. A large-canyon tree area in the canyon with a high H/W aspect ratio may adversely affect the tall building thermal load. Street albedo, enhanced by the color of tree leaves, can multiply the radiation reflections reaching the upper reaches of the canyon [242].

### 4.5. Design Parameters of Neighborhood Cluster Scale

As the scale increases, the relationship complexity between the elements that make up urban areas increases. At this hierarchy level, the high synergy that characterizes the ecosystem is noticeable. Allelopathic compounds, both antagonistic and non-antagonistic, can be found here. Understanding the urban unit as an interconnected network of abiotic and biotic objects allows us to grasp how the areas of coverage, density, and spatial configuration affect its temperatures.

#### 4.5.1. Coverage Area

Covering with elements of the urbanized environment is one of the indicators of the intensity of development, showing a close relationship with the surface temperatures [289,290]. Impermeable surfaces in the neighborhood unit represented by buildings and roads have constant high LST and UHI [161,172], and modulation of the coverage parameter strongly influences their variability [291,292,293,294]. The increase of paved surface coverage leads to LST and UHI increases [295]. Increasing the built-up area coverage by 1% may cause a rise of the UHI effect intensity up to 1.7% [170]. On the other hand, a 1% reduction may cause a decrease in the night-time near-surface air temperature by 0.1 °C [292]. Urbanization degree is proportional to the distance from the city center [187] and size-dependent urban area [172]. It is also inversely proportional to the greenery area [293,296]. Therefore, it seems logical that old cities are usually characterized by the highest concentration of impervious surfaces [136]. Large-scale architecture and infrastructure and a high percentage of commercial and industrial areas [122,170] have a high UHI correlation. It is worth mentioning that the modernistic districts from the 1950s–1960s are better in terms of the number of porous surfaces than the currently built districts and those from earlier centuries [234]. This is related to the Athens Charter implementation.

The building cover ratio in a given area significantly impacts the intensity of AT, LST, UHI, and PET [142,297]. As it increases, the degree of pedestrian-level ventilation decreases [298], and the LST and UHI intensity increase [299]. In China, with a 10% increase in building land cover within a 500 m radius, a daily T*_max_* increase of approximately 3.4 °C was noted [186]. The correlation of building cover with temperature can be positive and negative [299] and may similarly affect the nighttime and daytime UHI [180,204]. These relations are distinguished by the type of land cover, building materials’ thermal properties, and exposure of the urbanized environment to solar radiation [161]. Increasing the building cover degree may reduce the maximum T*_mrt_* during the daytime by reducing the canyon’s solar exposure [125]. A stronger positive correlation between building cover and temperature is noticed at night. That correlation results from the heat release after dusk from the housing estate thermal mass [294,300] and the anthropogenic heat trapped in the canyons [143]. The time at which the highest correlation is achieved depends on the temperature level on a given day and occurs later in the nights after hot days compared to the nights after cool days [238]. The influence of land cover on air temperature strongly correlates with built-up area density [129]. In a dense context, the percentage of built-up coverage has a greater impact on the increase in UHI than its density itself [250]. In this context, reducing buildings’ land cover would increase solar exposure for buildings and paved surfaces and thus affects the radiation and temperature increase [301]. However, paradoxically reducing the temperature in densely built-up areas is possible with a building cover increase [190]. There is, however, a limitation. The adjustable LST range decreases when the built-up area exceeds a certain critical threshold. This means the ability to mitigate high temperatures by adjusting the built-up area is also limited [182]. This ability is evidenced by the fact that, in some specific cases, increasing the land cover index in high-density regions may increase the temperature and UHI [129]. A possible reason for this is the reduced convection cooling at the pedestrian level [298]. In such a situation, it is necessary to reduce the coverage density to mitigate the UHI [129]. These dependencies may explain the lack of correlation between the building cover percentage and the UHI intensity in some studies [251].

The road coverage degree strongly correlates with the AT, UHI, and PET intensity [186]. Its increase within the city leads to a rise in LST [182,237] and enhances the UHI effect [185]. Due to the increased thermal mass, the duration of high air temperature is also extended [121], making the city’s microclimatic conditions more stable throughout the day. Unfortunately, this also contributes to the formation of the nighttime UHI. An increase in the road surface ratio causes a temperature increase [204]. That is especially true when the roads are highly congested—this increases the anthropogenic heat in the area [184].

Green area coverage is a significant predictor of elevated temperature for the neighboring unit scale [255], explaining up to 50% of the variability of intra-city temperatures [186]. There is a strong negative linear correlation between greenery coverage and the intensity of AT, UHI, PET [214,302,303], LST [266], and buildings’ peak cooling loads [224]. There is also a positive correlation with soil permeability, urban spaces water capacity, ground surface shade percentage, and wind speed [168]. The surfaces may have a similar daily effect on the UHI [161], significantly reducing the peak air temperature and the UHI during the day [302,304] and at night [305]. In a study conducted in China, a 10% increase in green area coverage decreased the mean UHI by approximately 0.94 °C in a 250 m radius [186]. Green area coverage shows a positive relationship with the SVF. Increasing the exposure of green areas to direct solar compensation results in better plant conditions and greater efficiency of evaporative cooling. Due to the above, these cover types cool the space exceptionally well in summer and in warmer climate types [185].

The water object coverage percentage shows a strong negative linear correlation with the LST, AT, UHI, and PET intensity [186]. The cooling effect of water reservoirs also depends on their size, location, and wind direction [138]. For example, after reducing the lake’s surface in the built-up area of Wuhan by 130 km^2^, the UHI intensity increased by 0.2 °C–0.4 °C [306].

#### 4.5.2. Elements’ Density

The density of the urbanized area is an urban geometry parameter that significantly affects the minimum temperature, LST, and UHI intensity [238]. This parameter is the most complex, as increasing the built-up area’s density may have a negative [190] or a positive impact on the LST and UHI [178,179,180]—depending on the climate type. Its importance is greater [197] in warmer regions than in cooler areas [190]. The urbanized area’s density is a hybrid parameter that describes the mutual synergistic relationship of other parameters, such as population density, anthropogenic heat, the vegetation ratio, building and road coverage, building height, and spatial configuration [118,155]. There is also a group of parameters such as the floor area ratio (FAR) related to density but primarily determining the building development volume. They significantly impact the thermal building mass and the shade patterns [155]. These parameters also negatively correlate with the LST and AT [307]. The influence of these aspects makes it challenging to infer one universal relation. The collective effect of all the elements influencing the density is the final density effect on the city temperatures [155,303].

The buildings and roads coverage density are mainly responsible for the density of land development. The influence of the building cover density on the ambient temperature rises with the increase of the built-up space radius [294]. The density is negatively correlated with the distance from the urban center [187]. The built-up area density parameter significantly influences the solar exposure of the housing estate [241]. As the built-up area density increases, the average SVF decreases [308]. A similar relationship occurs for the building’s volume [130], but it has a more substantial effect in already densely built-up areas [240]. The influence of these parameters on the air temperature is significant, but its distribution shows diurnal differences [302]. An increase in the buildings’ cover density and their volume can reduce the AT [241,304], the surface temperature of the lower canyon parts [152,217], and the heat stress at the pedestrian and UHI levels during the sunny hours of the day [183,304]. Those are easy to see in the example of squares. Their temperature can be even 18 °C higher than that of dense built-up areas [301]. During a cloudy day, densely built-up areas remain warmer, but the difference in T*_mrt_* between dense and scarce contexts decreases because of the reduced radiation intensity [301]. The increase of built-up areas’ volume and density can reduce the wind speed and the convective heat transfer. That increases the thermal mass, thus having a more substantial impact on the radial energy absorption and lower SVF leading to the thermal trap effect [235,241]. That leads to the growth of wall, street, and air temperatures in canyons and causes the nighttime UHI [227,241]. These dependencies are appropriate for warm seasons. In colder seasons, a temperature rise associated with high built-up areas’ density may be desirable to provide better thermal comfort [119,123]. The density is related to the anthropogenic heat emission [155]. A higher proportion of building walls means more anthropogenic heat and significantly influences the LST and UHI [170,180]. Depending on the season and context, it may positively or negatively affect thermal comfort and the environment. The increase in the built-up areas’ density reduces the annual energy demand [249,309] but makes the heating load more sensitive to solar radiation [238,307]. Moreover, intensified anthropogenic heat emitted from buildings may modify the thermal response of other city geometry parameters [310].

There is a particular paradox for the built-up areas’ density parameter—the total area of roofs, facades, and streets changes with an increase or a decrease of those areas [126,217]. That means one can increase the surface temperature by increasing or decreasing the building density. By increasing the density, we increase the fractional area of the roofs. As roofs get hotter more easily than streets, the UHI increases as the fractional roof cover increases [126]. On the other hand, we increase the sun exposure of road surfaces by reducing the density. Since roads are associated with a low albedo, they strongly influence UHI [152,244]. The solution to this situation is to manipulate the density by changing the height. In this way, it is possible to increase the shadow area without changing the fractional roof cover [126] and reduce the surface temperature as a result [244]. At a constant high density, the spatial configuration of blocks has a dominant effect on surface temperatures [179]. The increase in density for interlaced and linear block systems reduces the daily UHI [150], and the simultaneous increase in height and building density increases the frontal area index (FAI) [123,298] and night UHI effect [139]. This relationship can be seen in Hong Kong, where a 10% density and height reduction resulted in a 10% UHI reduction [303]. This shows how the density and configuration of construction objects on the estate are strongly correlated.

The road surface density depends on the urbanization degree of the area [296]. The LST and UHI are positively correlated with the coverage density of roads, parking lots, and sidewalks [311,312]. This is directly influenced by the thermophysical properties of the materials used for the construction [170]. However, the road connections density alone is insufficient to understand the relationship between city texture and the UHI effect. Other factors, such as road connection size and capacity, are also important determinants of the UHI effect intensity [170]. The areas closer to the functional city center are characterized by more significant node and communication link density, making these areas warmer than the rest [185,189]. The UHI effect is negatively affected by a higher transit route density. The reason may be trivial—faster vehicle traffic and the absence of congestion reduce the amount of heat, pollutants, and greenhouse gases discharged into the canyon [170]. Scattered and anisometric settlements are associated with more traffic, affecting higher emissions [291,313]. Therefore, increasing the built-up areas’ density can help deal with anthropogenic heat emission problems [170].

The vegetation density is determined by its height, cover type, and greenery amount. It depends on the growing season during a year, too. In addition to shade patterns, the density of the vegetation models the airflow [235,314]. Increasing the greenery density within a green area reduces not only the daily LST [171,172] but also the wind speed [235,314] and the convective cooling effect. Increasing the vegetation patch density reduces temperature [190,295], especially at night [150,273]. Increasing the edge density of these patches reduces the ambient LST by having a larger heatsink contact area with the paved area and increased energy flow between them [190]. The vegetation cover density is negatively correlated with the built-up areas’ density [155,180]. The more dense built-up area ensures greater efficiency of green roofs and green facades [183]. An increase in built-up area density increases green infrastructure’s impact on the LST changes [179]. However, after exceeding a certain density threshold, other variables related to the housing estate spatial configuration, e.g., the heights of buildings, begin to significantly impact the lowering of temperature by vegetation [310].

#### 4.5.3. Spatial Pattern

The 2D and 3D composition of the housing estate, which consists of the mutual spatial relationship and the configuration of green areas, water objects, streets, and blocks of flats, is significantly related to the LST and UHI [202,208].

The spatial distribution of the impervious surface within the estate, significantly when fragmented and insulated, can increase the LST [123]. On the other hand, increasing hardened surfaces with complex shapes sometimes cools the study area because of a more extensive contact between built-up areas and vegetation [190]. However, the relationship emergence condition is the presence of green spaces in the vicinity. The permeable surface system of green areas may play an essential role during intense solar radiation [54]. A compact green-space layout is generally accompanied by more significant amounts of permeable land [234]. An evenly distributed complex shape pattern of trees and its high dispersion can provide more shade and enhance energy exchange between green spaces and built-up areas while reducing the LST [190,234]. Moreover, low plants’ scattered distribution and irregular shapes can help mitigate the UHI effect [122]. In the case of water objects located within the estate, their cooling effect depends on their location in the urbanized area and wind directions [138].

The layout of buildings may be the most influential factor in changing the thermal environment during the day [54]. Different urban designs affect the insolation and ventilation patterns of the housing estate diversely [315] and thus determine their thermal efficiency during the day and at night. Clustered blocks of flats raise the AT and LST [121,122] because of the higher roof surface density and a lower greenery percentage. Therefore, the dispersed distribution of buildings facilitates the LST reduction [122], allowing the introduction of other mitigation features into the estate. It has been proven that the surface temperature and thermal comfort outside free-standing blocks of flats are slightly influenced by street orientation and H/W. The vegetation has a stronger influence [208]. Thus, in the absence of sufficient greenery, a higher open-space factor increases the UHI effect [255,312] and T*_mrt_* because of increased direct sunlight exposure [281]. Regular block of flats patterns may facilitate air circulation in the canyon [170]. Higher irregularity in the layout of buildings, and the complexity of their shapes and diversity, affects wind patterns and also increases LST and UHI levels through higher solar exposure [190,301].

The dense linear apartment block systems that make up the city canyon have the highest UHI level [316] because they provide a long solar radiation duration for the estate [166,203] in the case of incorrect spatial orientation. By controlling the direction of the linear form, it is possible to influence both passive cooling and heating of the blocks of flats [166]. This way, the apartment block orientation as a whole has a significant impact on energy consumption, especially for high linear systems [309]. The windward buildings are blocking the airflow [274]. Therefore, adjusting the building blocks’ layout and orientation concerning the prevailing wind directions is essential. Proper arrangement of buildings can positively affect urban porosity, create wind corridors, strengthen the narrowing effect in the ventilation corridor, and increase wind speed [274,298] and thus can shorten the duration of high air temperatures in the area [176]. On the other hand, inappropriate building arrangements and orientation, increased building density, asymmetry, and the number of alleys may block the airflow, resulting in increased temperature at night [131,274]. For example, suppose the binary system is perpendicular to the wind directions. In that case, its ventilation and thermal comfort in summer can be improved by increasing the space between buildings along the road and creating small green areas [253,298]. However, this treatment will lower thermal comfort in winter [253]. An example of traditional Greek city development shows how to eliminate this problem by using small linear blocks of flats and a complicated street pattern [132].

Although building arrangements may negatively affect air circulation, they provide more shade [274]. Therefore, the closed layout of the apartment blocks is characterized by a more comfortable thermal environment during the day [274,309], provided that the H/W ratio is high. However, the low SVF associated with this system may result in poor radiant cooling capacity and thus increase the nighttime temperature [305,306]. The low H/W ratio of the closed apartment blocks layout increases its LST [305]. Nevertheless, the natural compactness of the apartment blocks with a peripheral configuration makes these systems more suitable in an urban environment than linear systems [150].

The average building height significantly affects the ambient temperature [157,197] and wind patterns [203,241]. It also determines the shading value in densely populated urban areas [298,301,308]. Therefore, greater building heights indicate a lower AT and a better outdoor comfort level [217,302]. Building height can influence cooling efficiency through its relationship to the UHI formation and propagation. Up to a certain height, the building’s height strongly influences the evapotranspiration cooling and the shade provision by the vegetation in the estate [178]. When buildings are lower than trees, the buildings do not obstruct the cool horizontal airflow and do not reduce the cooling capacity [152,275]. As the buildings’ heights increase, the average wind speed above buildings increases [265]. However, it harms the ventilation at the pedestrian level [275] and increases the internal heat gain by thermal trapping [139]—especially in public and residential areas [122]. The buildings’ heights increase the urban unit’s FAI, thus blocking the natural wind corridors [166]. Due to all these unfavorable relationships in the case of high buildings, the influence of ventilation significance on city air temperatures rises [44]. It is worth introducing a gradual lowering of buildings in the direction of prevailing winds to reduce the FAI [166]. It is also possible to modulate the urban roughness, determined by the differences in the buildings’ heights. This affects both the shadow distribution and the wind speed and direction in urban areas [176]. Hence, it may also impact the UHI effect [126,197]. Identical building heights lead to a stronger vertical turbulent movement favorable for pedestrian-level ventilation [275] and increase the albedo [217]. However, an increase in the difference in building height increases the shadow area and the aerodynamic roughness. That affects convective district cooling positively, especially at night [126]. It should be remembered that high surface roughness values make it challenging for adequate wind ventilation [176] through the vortices generated. It can be concluded that the benefits of this parameter modulation in an urban environment can be obtained by carefully mixing the building heights [150,166].

For the elongated apartment blocks’ shapes and increased H/W ratios, block orientation becomes a vital parameter influencing temperature rise and energy consumption [239,261,309]. Buildings facing south have the highest daytime temperatures for the northern hemisphere because of insolation [274]. Although these are suitable conditions for a low-energy urban form, this solution’s balance of losses and profits loses to other savings forms [309].

## 5. Conclusions and Recommendations for Future Research

The literature review revealed significant urban environment parameters in terms of the UHI, which was discussed in the above section. The research specified three groups of parameters corresponding to their geometry, morphology, and topographic relation. The parameters significantly affecting the city’s temperature are: (1) building density, urban surface albedo, and distance from the city center for NH; (2) aspect ratio, ground surface albedo, and street orientation for SC; and (3) building height, material albedo, and building orientation for BU. The diversified distribution of research interest is presented in various examples from different countries. Moreover, the relationship between the BGI [92] and UE parameters was confirmed. The BGI and urbanized environment parameters facilities overlap in some dependencies. Significant geometrical urban space parameters turn out to be critical topographic parameters for the BGI facilities. That confirms the correctness of understanding the greenery and water objects’ impact on the city geometry UHI mitigation, and thus the synergy effect in the city.

Based on the analysis in the discussion chapter, it was possible to create microclimatic design guidelines at various scales.

### 5.1. Guidelines for Microclimatic City Design

The complete elimination of the UHI effect seems impossible in the current urbanization form. However, it is possible to improve the thermal city environment. Following some universal guidelines to improve the existing urban tissue and create the new one makes it possible to influence the entire city PET and UHI effect significantly. Only holistically planned preventive measures and interventions can make the city climate-neutral and make it resistant to the rapidly approaching climate change.

The guidelines based on the discussion chapter show a strong shifting vector of changing the current urban environment toward de-urbanization. This rather old concept returns in the climate change context of the 21st century and sheds a new “green light” on the city’s appearance in the future. As in the de-urbanization concept, the microclimatic design reduces and distributes rationally the built-up areas’ coverage and the density and height of the buildings, concerning the natural landscape and well-known weather patterns. The main principle of this design trend is to adjust the solar intensity as the leading energy carrier to the needs of the specific temperate climate type. In warmer varieties, less radiation will be needed to provide the same conditions as in colder regions. When planning the radiation balance in the urban areas, one cannot forget the vegetation duration appropriate for each space. It is necessary to consider both in situ and ex situ factors in microclimatic design because the city’s energy flow remains open [151]. The designers should not rely on wind cooling a city with a poor wind environment and avoid creating large and compact urbanized areas [313]. They should also intertwine building areas with forests and lakes, develop urbanized areas toward greenery, and create decentralized systems with highly amorphous, organic shapes. Inside the designed cities, they should diversify the environment, improve the distribution of buildings, roads, and BGI elements [168,172], reduce the amount of industrial and commercial areas, limit road transport (by planning ring roads and smart transport links), and promote a zero-emission life [90].

#### 5.1.1. Material

When creating, introducing, or modifying city materials, one should pay particular attention to some of their properties. It is recommended to use materials with increased diffusivity, transmission, and a reduced speed of radiation absorption [199]. It is necessary to increase pavement permeability and the materials’ porosity [206]. Permeability can be increased by using various types of vegetation surfaces. Cities’ poor ventilation requires the extension of road material porosity [182]. When night ventilation is provided at a high level, materials with a high heat capacity, such as grass, stabilized sand, and granite, can be used [193]. In any other case, it is recommended to reduce the heat capacity and thermal mass using, for example, reclaimed asphalt pavement (RAP) materials [206] or limiting the thickness of the foundation layers [199]. The use of asphalt and concrete as a surface is strongly discouraged [193]. Thermal insulation of building partitions should be increased, especially in colder climate zones.

The urban areas’ albedo should be varied [188]. Retroreflective materials can be safely used on all hard surfaces and contexts [317,318,319]. The guidelines for high albedo materials are dependent on the area density. In the densely built-up areas, the albedo should not be increased. In this situation, attention should also be paid to the leaf’s albedo. It is safe to use high albedo materials only on the highest roofs. Increasing the albedo on each surface for low H/W ratio values is desirable. When increasing the albedo, the PCM and RR coatings, white paints and membranes, bituminous surfaces modified with oxides, colored roof tiles [320], and resin surfaces with exposed light-colored aggregate can be used [198]. It is necessary to ensure high color durability. Resin surfaces have significant limitations in use. It is recommended to use them as pavement surfaces for pedestrian use [211].

#### 5.1.2. Building

In the case of buildings’ geometries, it is necessary to reduce both their floor and wall areas. This is essential for commercial and residential buildings. Facilities must take on a compact, regular, and simple form [240]. The best solution would be to use spherical, domed shapes [321]. In addition, the buildings’ heights should not be significant. It is suggested their heights should be in the range of 20–50 m [265] and much lower for buildings located near main roads [294]. The best solution is to place buildings on high pillars [298].

Increasing the buildings’ energy efficiency and reducing their thermal emissions is necessary [197,247]. For this purpose, one can use location-related, orientation, greenery, building covering materials, and cooling or heating solutions. One should also avoid using the A/C. If this is not possible, it is recommended to use A/C and heating in energy-saving mode and timers [135]. Instead, it is possible to use ground heat exchangers and water cooling [177]. It is also worth using other low-energy solutions. For example, RR materials are desirable to limit A/C usage [232]. A similar task is fulfilled by increasing the building albedo [197,224] and applying better insulation [197]. Green roofs effectively reduce cooling and heating loads [141,178]. In addition, buildings should be reasonably located in relation to chiaroscuro and wind corridors, depending on the desired effect.

#### 5.1.3. Street Canyon

Proportions, orientation, and greenery patterns should be selected appropriately [208,268]. The designer’s primary goal should be to obtain a low solar exposure of the canyon [322], but the optimal value for a given climatic zone should be considered [256,278].

The most crucial guideline for canyon geometry is to reduce its height while increasing the tree crown cover [278,323]. In hotter types of climate where tree shade cannot be introduced, canyon height can be increased [258]. Limiting the canyons’ heights is especially important in the case of canyons that are transverse to the main winds. In this case, it is worth increasing a building’s height slightly on the leeward side and increasing the spacing between the buildings along the road. The depth of the street canyon oriented parallel to the airflow may be higher, but it is worth increasing their horizontal asymmetry and smoothness. Nevertheless, it is worth reducing the heights of buildings in the vicinity of intersections. Reducing the road surface area or increasing shading solutions is another critical issue. The answer is to use traffic circles, reducing traffic congestion and the anthropogenic heat increase.

There is no single preferred city canyon orientation. The desired direction varies with the climate zone and the need for sunlight or shade [155]. The best general solution seems to be the use of irregular road runs, in which the orientation changes from time to time, creating a balanced thermal environment. In addition, it is worth using diagonal streets [264,324]. In warmer temperate climates, the orientation of the roads along the N–S axis is suggested as the most appropriate [253,281]. In this situation, it is worth ensuring a considerable height of the canyon [222,324]. One can also reasonably increase the streets’ width to provide better comfort of their use [322]. It is also good to extend the road length unless it is an arcaded street. Streets oriented on the E–W axis are suggested as optimal for colder regions [285]. They may also be optimal in warmer climate zones, but only if the canyon is deep [283,324]. In that case, the canyon’s length should be shortened [250], and the spaces between buildings along the street should be increased.

Shaders (artificial or natural) in canyon space are generally considered a good solution [262,269]. A tree crown cover is preferable to artificial shaders for many ecological and microclimatical reasons [136]. The trees should be as tall as possible to create a lot of shade and ultimately cover most of the street. However, their use legitimacy should be considered, and the optimal tree species should be selected for the given canyon openness and orientation [281,287]. It is necessary to introduce trees into vast canyons [251,283]. In wide canyons, dense canopy cover and compact artificial shaders should be used. It is not recommended to introduce many trees into high canyons, and their canopy cover should be sparse [135,286]. Shading with greenery, arcades, or artificial shaders should be used to avoid the E–W-oriented streets overheating [259,283].

#### 5.1.4. Neighborhood

During the construction of new neighborhood units, it is recommended to generally reduce the intensity of urban development [299] and manipulate the parameters of the spatial configuration to create the most diverse landscape. In this way, certain areas will lower daytime temperatures, and others will reduce nighttime temperatures, and the whole will ensure correct cross-ventilation [151]. It might seem the best development strategy for an urbanized area for its microclimate is the foundation of skyscrapers surrounded by greenery [325]. However, the efficient creation of an optimal thermal environment is also possible in other already existing conditions.

The main goal of microclimatic design on the city scale is to reduce the urban surfaces cover (impermeable) in favor of natural ones (pervious surfaces). The buildings and roads should be reduced, thus increasing the fraction of the green spaces and water object areas [295,300]. That is especially important in the context of very intensively built-up areas [250,325]. It is sometimes enough to decanalize rivers hidden underground during intense urbanization to increase the number of water objects.

Reducing the road and built-up area densities and volumes to increase the BGI coverage density is generally required to improve the urban climate [300,326]. The legitimacy of modulating the built-up area density depends on the climate type and the possibility of introducing tall greenery. For this reason, in warmer temperate climate zones, it will be justified to increase the built-up area density to ensure better thermal comfort in the urbanized area [256]. The transit road density may be increased, but only when accompanied by shading trees. It is not recommended to excessively increase the tree-planting density in green areas; instead, tree species with the desired parameters should be selected appropriately for the available space [234].

One should reasonably manipulate the area density by distributing the buildings horizontally and vertically [278]. A desirable solution is to lower the buildings’ heights and carefully differentiate the buildings’ sizes in apartment block areas [166,278], especially those with high building density [129,176]. In that case, the climate variation must be considered. In warmer regions, it will be preferable to increase the buildings’ heights to increase density [122] and decrease the heights of south-facing apartment blocks. Moreover, to maintain the same built-up areas density, it is possible to reduce the coverage area with low buildings in favor of tall buildings [172]. An essential aspect of creating a housing estate microclimate is to ensure even wind access throughout its area [176]. For this purpose, tall buildings must not be located on the neighborhood units’ outskirts from the prevailing wind’s side [166]. The same principle applies to tall buildings near water objects and green areas. The buildings’ heights may increase with the distance from the boundary and cooling objects. Nevertheless, it is worth adjusting the heights of the tallest buildings so that the roofs are under the tree crowns. The designers should modulate the buildings’ sizes to allow the wind and humid airflow to penetrate the housing estate’s interior [294]. They should plan uniform and compact green areas that are spatially interconnected [234]. In high-density built-up areas, one should create irregular green areas and fill any free space with them [182]. Green roofs and facades are recommended [142]. Even distribution of the tree crowns in the housing estate area should cover as much area as possible [54,294]. Water objects should be located from the windward side. It is crucial to not canalize streams and rivers on a given site [294].

The relationships of the shape, length of apartment blocks, and spaces between buildings should be rationally planned, considering the creation of insolation areas and wind corridors [274]. The best buildings’ covering pattern is dispersed distribution in the neighborhood greenery [54,182]. This principle also applies to the housing estate layout design [294], especially if the surroundings are excessively dense built-up areas [121]. It may also be helpful to use a variable apartment building configuration [121]. One should avoid building an estate as a closed structure of excessive depth [54]. Instead, it is better to use average building heights, which let the sunlight reach the bottom of the courtyard. Furthermore, designers should orient regular patterns relatively parallel to the prevailing wind directions while controlling the rotation relative to the optimal solar exhibition. Finally, free spaces between buildings appropriately oriented to the wind corridors are recommended.

### 5.2. Vision of Climate-Resilient and Climate-Neutral Cities

The urbanization principles need to be changed to eliminate the UHI effect and thus ensure optimal PET comfort in the city. In addition, the microclimatic approach to housing estate design may affect the scale of the changes. Still, the vision of a future temperature increase in a temperate climate zone requires researchers and designers to brainstorm and create ideas for preventing a simulated crisis.

According to the ecosystem pulse theory, each civilization’s violent disintegration is associated with an excessive complexity increase [327]. Research clearly shows that urban development and uncontrolled spatial configuration exacerbate the temperature difference between urban and non-urbanized areas. Vegetation is essential to achieving ecosystem climax, and progressive surface reduction leads to climate anomalies and crisis deepening.

In the contemporary understanding of urban planning, solar exposure is the most critical factor influencing the overheating of urbanized areas. Of course, there are more reasons. Still, using already known smaller-scale solutions, solar exposure can be transformed into something desirable together with the underground placement of the building. With this change of approach, the problem becomes the solution. Earthship-type buildings show that construction can be energy-saving and reduce emissions to virtually zero. In that case, the large thermal mass of buildings also becomes an advantage, and its internal microclimate is easier to control because of the constant lower temperatures prevailing underground [328]. Despite the modern name, life underground does not bear the hallmarks of modern times. Since prehistory, people worldwide have lived in earthworks [329,330] and archaeological research shows that even entire cities can be created this way [331].

The vision of the underground city seems abstract, but only in this way is it possible to ensure a parallel, undisturbed existence of nature and architecture. Building cities underground does not seem to be technically impossible, and that concept appears from time to time in urban planning and architecture [332]. The earthscraper is a skyscraper, only upside-down—instead of rising, the earthscraper is built deep into the ground. Building heating requirements will force designers working in the field of temperate climate area to change the apartment blocks’ spatial patterns, taking inspiration from warmer climates [203,333]. Creating neighborhoods covered with the earthscrapers and Earthship-type buildings surrounded by greenery, water features, and underground public transport can effectively solve the upcoming climate challenge.

### 5.3. Future Research Direction

Based on this review article, three main directions for future research can be identified: (1) carrying out a similar analysis for different climate zones; (2) implementing and testing the developed guidelines in different urban surroundings; (3) theoretical work and practical tests related to developing a vision of underground urbanization; and (4) developing a clear range of possible solutions and materials for both BGI and the UE.
Reviews of the UE parameters relevant to the UHI propagation for different climate zones are necessary to develop homogeneous guidelines for every climate zone separately. Only then would it be possible to compare the microclimatic design policies and learn about the diversity of the individual parameter impact in different climate zones. The model of this publication can be used to create a normative series of reviews.The introduction of changes to the urban environment to modify the microclimate is already taking place. Microclimate modeling programs (such as the ENVI-met) implemented geometry changes and UE morphology solutions. BGI implementation is carried out as well. Although these programs are based on many years of research, it is impossible to generate solutions depending on the preferred temperature change automatically. The research on an automated algorithm able to model a city’s thermal solutions will be the area of the next stage of research of the presented paper’s authors.Another research area worth undertaking is underground building development. Models should be made based on the theoretical knowledge obtained from this review, creating prototype constructions and analyzing their performance. Solutions should be designed to enable the entire urban unit’s autonomy in energy and a closed cycle of matter and raw materials. The review authors plan to research these solutions in the future.The necessary gap to fill is the development of a clear range of techniques and materials desirable for microclimatic reasons, which can be used in local interventions and in creating a new urban environment. A good solution would be to construct specialized tables for both BGI elements and UE facilities listing possible techniques, solutions, and plant and building material pallets. Material pallets should define their averaged quantitative impact on the microclimate and the urban environment quality, including carbon dioxide, oxygen, temperature, humidity, dust, wind, and shade. Full parameterization of the solutions range should form the basis for further research into microclimatic design automation.

The results of this work should serve as the basis for a broad discussion on the architectural and urban environment. Based on the results of the review, in consultation with local governments, strategies for adaptation to climate change and plans for the development of new urban forms may be developed. The authors declare actions for the future verification of the developed design guidelines. Appropriate steps will be taken to popularize the achieved results among professionals, local authorities, and researchers in the form of training sessions, courses, and industry articles.

## Figures and Tables

**Figure 1 ijerph-19-04365-f001:**
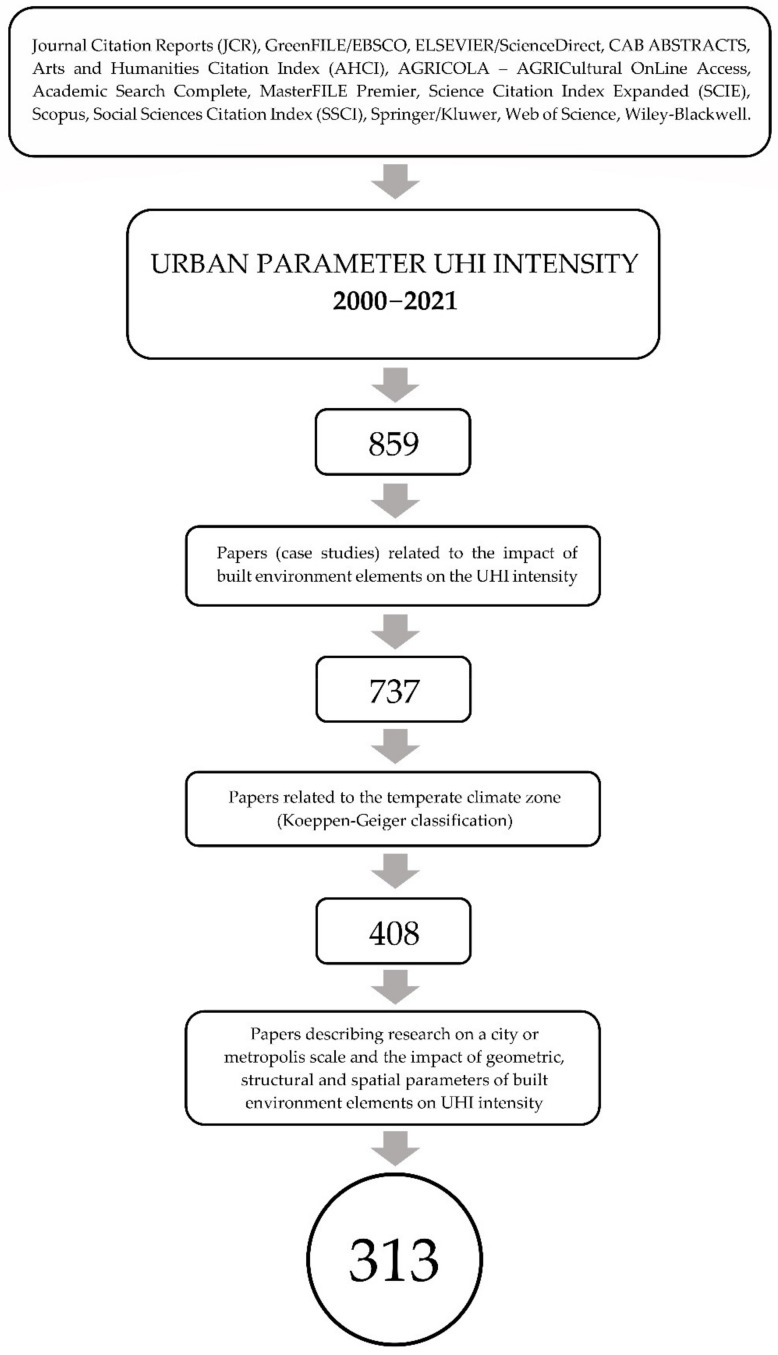
Logical diagram of papers selected for the research.

**Figure 2 ijerph-19-04365-f002:**
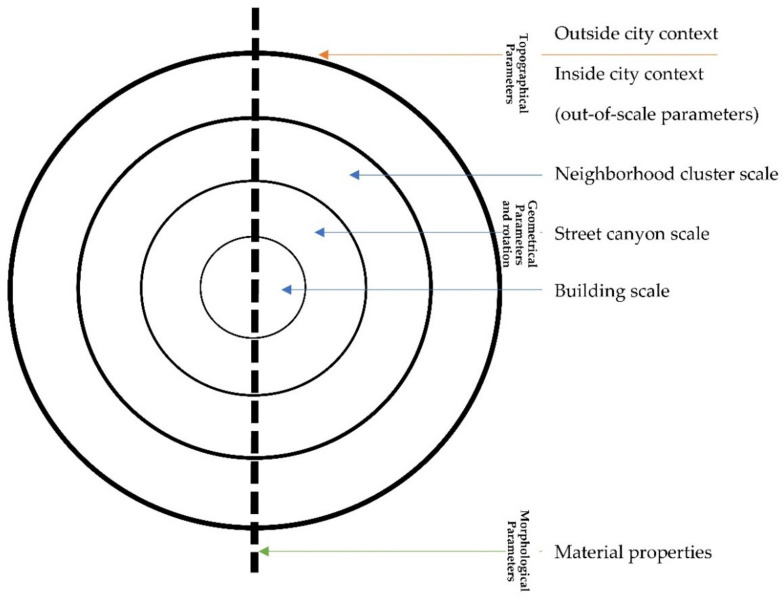
Logical diagram of the hierarchy and relationships between scales and parameters’ families.

**Figure 3 ijerph-19-04365-f003:**
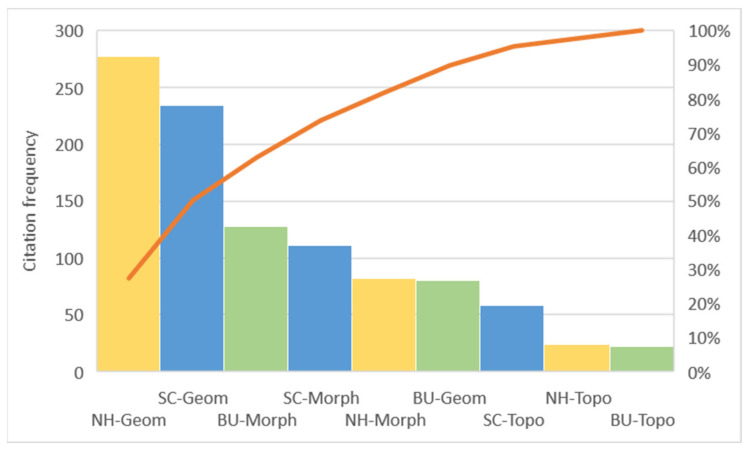
Pareto diagram. The frequency of scientific paper citations based on the analysis of the parameters’ families.

**Figure 4 ijerph-19-04365-f004:**
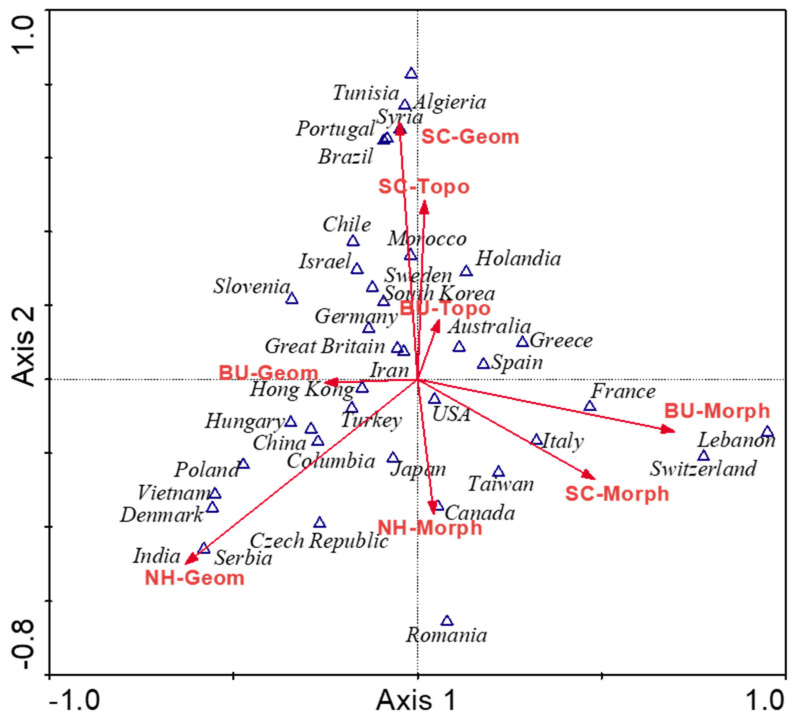
The canonical correlation analysis (CCA) (*n* = 948). Relationships between the research frequency and parameter families in selected countries based on papers included in the analyses.

**Table 1 ijerph-19-04365-t001:** The review of the blue-green infrastructure (BGI) element parameters relevant to urban heat island (UHI) effect mitigation [92].

BGI Structure	Parameter Family	Dominant Parameter
Water structure	Geometrical	Area
	Morphological	Degree of vegetation along the bank
	Topographical	Wind exposure (trend of cold transfer)
Green area	Geometrical	Area
	Morphological	Percentage of an area covered by trees
	Topographical	Exposure to solar radiation/degree of shading by surrounding structures
Greenery along the street	Geometrical	Tree crown width/diameter
	Morphological	Leaf size/leaf area index (LAI)
	Topographical	Canyon geometry/height/width
Green roof	Geometrical	Substrate layer thickness
	Morphological	Degree of hydration/moisture of the substrate
	Topographical	Height of structure above the ground/distance from the nearest BGI (synergy)
Green wall	Geometrical	Degree of vegetation coverage of a building/the extent of the green wall
	Morphological	Leaf width/leaf area/foliage density/LAI
	Topographical	Orientation relative to the sun

**Table 2 ijerph-19-04365-t002:** Review articles published so far on the impact of urbanized environment (UE) parameters on UHI propagation.

Author	Title	Usefulness for Creating Design Guidelines
[93]	Street geometry factors influence urban microclimate in tropical coastal cities: a review	Identify the impact of street geometric, morphologic, and topographic factors and explain water body effects on urban microclimate. The study was conducted only for tropical coastal cities.
[94]	Urban design parameters for heat mitigation in tropics	Focus on heat mitigation strategies, with the use of modifications and urban geometry (shading), urban greening (street greenery, parks, green walls, green roofs), urban ventilation (street orientation), albedo, and water bodies area. Case studies only from the tropics.
[95]	Street design and urban microclimate: analyzing the effects of street geometry and orientation on airflow and solar access in urban canyons	Discussion about street geometry factors and street topography (orientation), mainly focusing on parameters providing solar access and airflow along streets. Hot climate regions case studies.
[96]	The impact of urban design descriptors on outdoor thermal environment: a literature review	Design strategies based on parameters: canyon geometry and topography, land-use intensity, building form and materials, space enclosure, and urban vegetation. The research focused on street scale urban thermal comfort. In addition, it considered the synergy of most climate zones case studies.
[97]	A parametric approach to optimizing urban form, energy balance, and environmental quality: the case of Mediterranean districts	The paper lists building and street form design parameters: shape factor, floor area ratio (FAR), site coverage, orientation, sky view factor (SVF), aspect ratio (H/W), the distance between buildings, and average building height. Case studies from different climate zones.
[98]	Review of the impact of urban block form on thermal performance, solar access, and ventilation	The research discussed urban blocks’ geometrical parameters impact and buildings’ geometrical and topographical parameters. It is focused on solar access and ventilation. Case studies from different climate zones.
[31]	A review on the generation, determination, and mitigation of the UHI	List case studies for UHI mitigation strategies based on urban matter parameters alteration: albedo, air conditioner occurrence, building design, water bodies. Case studies from different climate zones.
[99]	A review on outdoor thermal comfort evaluation for building arrangement parameters	Building Arrangement Parameters for the densely built area impact the outdoor thermal comfort. Parameters listed in a review include SVF, H/W, building height, plot size, pavement cover ratio, green plot ratio, vehicle traffic density, building density. Cases from different climate zones.
[100]	Overview of the UHI phenomenon toward human thermal comfort	Lists building and street design parameters such as albedo, imperviousness, canyon openness, glazing ratio, H/W ratio, type of material, heat conductivity, orientation. There are examples from different climate zones.
[101]	The influence of building height variability on natural ventilation and neighbor buildings in dense urban areas	Focus only on the influence of building height and dense urban areas. Case studies form different climate zones.
[102]	Reducing urban heat island effects: a systematic review to achieve energy consumption balance	The paper considered different types of green spaces and material properties based on parameters (incl. albedo) as a design strategy for UHI mitigation.
[103]	Combating urban heat island effect: a review of reflective pavements and tree shading strategies	The strategies focused on applying chip seals, white toppings, and coatings were discussed. In addition, the role of surface reflectance, including those from asphalt and concrete pavements, albedo improvements and technological trends, application of waste materials, and industrial by-products, are presented. Finally, the contribution of urban tree shading systems to pavement temperature and microclimate systems is presented.
[104]	Thermal performance of cooling strategies for asphalt pavement: a state-of-the-art review	The authors show how alternating road material thermophysical properties throughout the material mix and treating changes can affect thermal comfort in cities. They describe the role of vegetation in cooling strategies. Case studies from different climate zones.
[105]	Urban heat island: causes, consequences, and mitigation measures with emphasis on reflective and permeable pavements	The paper focuses on reflective and permeable pavements and lists road and pavement morphologic parameters: material density, albedo, emissivity, heat capacity, thermal conductivity, thermal diffusivity, thermal admittance. Case studies from different climate zones.
[106]	Cool pavements for urban heat island mitigation: a synthetic review	Investigating the efficiency of reflective and permeable pavement as the UHI control elements. The USA case studies from different climate zones.
[107]	Sustainable pavement: a review on the usage of pavement as a mitigation strategy for UHI	Short description of the UHI mitigation strategies: green and blue surfaces, cool roofs, cool pavements (reflectivity, evaporation rate, permeability, porosity, water content, and heat conduction). Case studies from different climate zones.
[108]	Review of studies on outdoor thermal comfort in warm, humid climates: challenges of the informal urban fabric	The paper describes the parameters’ influence on outdoor thermal comfort: canyon axis orientation, SVF, H/W. Studies only for warm and humid climates.

**Table 3 ijerph-19-04365-t003:** The urbanized environment (UE) elements’ validity classification influencing the urban heat island (UHI) effect intensity in a temperate climate zone.

UE Structure	Parameter Family	Dominant Parameter	Number of Reports in Scientific Papers	Percentage
Neighborhood cluster (NH)			380	100
	Geometrical parameters		277	72.3
		Building density (BD)	76	-
	Morphological parameters		82	21.4
		Urban surface albedo (WAS)	24	-
	Topographical parameters		24	6.3
		Distance from the city center (CBD)	6	-
Street canyon (SC)			398	100
	Geometrical parameters		234	58.1
		Aspect ratio	88	-
	Morphological parameters		111	27.5
		Ground surface albedo	25	-
	Topographical parameters		58	14.4
		Street orientation	50	-
Building (BU)			230	100
	Geometrical parameters		80	34.8
		Building height (BH)	55	-
	Morphological parameters		128	55.7
		Material albedo	40	-
	Topographical parameters		22	9.5
		Building orientation (O)	12	-

**Table 4 ijerph-19-04365-t004:** Statistical parameters for the canonical correlation analysis (CCA) presented in Figure 3.

Number of Variables	9		
Number of Rejected Variables	1		
Number of Permutations	9999		
Parameter Family ^1^	*p*-Value	F-Value	% Expl.
NH-Geom	0.001	25.69	12.63
SC-Geom	0.001	21.08	10.22
BU-Morph	0.001	19.36	10.79
SC-Morph	0.002	15.23	8.66
SC-Topo	0.003	13.56	7.66
NH-Morph	0.005	10.49	6.15
BU-Geom	0.012	7.36	5.33
BU-Topo	0.028	5.88	5.07

^1^ NH-Geom–neighborhood cluster-geometrical parameters; NH-Morph–neighborhood cluster-morphological parameters; SC-Geom–street canyon-geometrical parameters; SC-Morph–street canyon-morphological parameters; SC-Topo–street canyon-topographical parameters; BU-Geom–building-geometrical parameters; BU-Morph–building-morphological parameters; BU-Topo–building-topographical parameters.

## Data Availability

Not applicable.

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
