# Peer review of "The Future of Climate-Resilient and Climate-Neutral City in the Temperate Climate Zone"

_ijerph, 2022, doi:10.3390/ijerph19074365_

Round 1

Reviewer 1 Report

The article is exhaustive and well structured. The reader would benefit from an annex of glossary of terms with the abbreviations used, to simplify the reading of the text and tables.

My only comment would be to enrich the prespective of  green as material. The selection of the vegetal palette in cities could not only contirbute to the shading for their canopy and morhopolgy, but also for their distict capacity of CO2 retention adn O2 production according to the species.

Nevertheless, I would like to congratulate the authors for an interesting and focused piece that article should be accepted in its present form on my opinion.

Author Response

Reply to review #1

1. The reader would benefit from an annex of glossary of terms with the abbreviations used, to simplify the reading of the text and tables.
Ad. 1. Following the journal recommendations, all abbreviations firstly used in the abstract, tables, figures, and the main text have been fully expanded.

2. My only comment would be to enrich the prespective of green as material. The selection of the vegetal palette in cities could not only contirbute to the shading for their canopy and morhopolgy, but also for their distict capacity of CO2 retention adn O2 production according to the species.
Ad. 2. As stated in section "1.2. Countermeasure strategies" under Table 1, there is a short note on the impact of BGI solutions on microclimate and urban air quality. The development of the topic is important to us and will constitute the next stage of creating the BGI modeling algorithm. Developing a material palette concerning greenery would be an integral part of the following article in the series. This direction is described in detail in point 4. in section "5.3. Future research direction".

Author Response

Reply to review #2

Main remarks:

1. Grammar and writing rules should be carefully checked.
Ad. 1. The grammar and spelling have been thoroughly tested. They have been corrected throughout the text.

2. For all the captions of Tables and Figures in this manuscript, more detailed explanations should be added without using abbreviation so that readers can understand the figures and tables per se not referring to the text.
Ad. 2. Abbreviations have been removed from the titles of tables and figures and replaced with full expansions. In the table contents, the first use of the acronym was correlated with its full name. Due to the table 4 character, its content has not been changed – abbreviations are described below the table. The abbreviations in Appendix 1. are associated with the full names. Additional acronyms appearing in this table are also associated with their full names, synonyms. They represent the input parameters for the review and are not its explicit content. The authors believe that their description is therefore sufficient.

3. The Title need to be concise
Ad. 3. The title was shortened by getting rid of its detailed extension. The current form is "The future of climate-resilient and climate-neutral city in the temperate climate zone".

Detailed comments:
1. Page 5 / Lines 108-109: I think the relationships between the urbanized environment parameters and the city's climate are interactive rather than dependent. Please explain this statement or revise it.
Ad. 1. We agree with the comment. This logical error has been corrected, and its present form is "The urbanized environment parameters influencing the temperatures inside the city are highly interactive with the city's climate".

2. Page 5 / Lines 109-110: I think climate is stemmed from the accumulation of long-term annual weather conditions. In other words, 'climate' is defined by 'weather', not the other way around. Please explain this statement or revise it.
Ad. 2. We agree with the remark. This logical error has been corrected, and its current form is "Climate describes annual geographical location weather patterns".

3. Overall: The authors classified the variables into 3 different groups: geometry, morphology, and topography. This scheme is logical and understandable to me, too. However, as the 3 families are not absolutely independent one another, it is needed to explain about how to address the conceptual overlaps between the families in the section of Introduction or Materials and Methods.
Ad. 3. A new section was created in the methodology chapter "2.2. The systematizing elements structure" to explain the scale hierarchies, parameter group distribution, and the interaction between scales and parameters. This section describes the individual scales and describes their correlations with each group of parameters in detail. The reasons for assigning particular elements to given taxonomic groups are described here. The penultimate and last paragraph of this section explains the problem of overlapping parameter groups and assigning them to a given scale. Figure 2 has been updated with the parameter groups distribution to illustrate those interactions better. We believe that we have sufficiently explained to the reader the logic of choosing a given methodology in this way.

Reviewer 3 Report

Dear authors, I really like how you set the scene in explaining in clear words the UHI and its effects, the temperature climate zone, etc. - well done!

This paper is a very good overview and summary of all different parameter categories affecting UHI.

Just a few minor things: please write PET explicitly as "Physiologal Equivalent Temperature", this is not been done throughout the whole document (and there might be readers who are not familiar with this). In general, it would be good to have a table of abbreviations at the end of the paper, as there are many.

For further research: it would be good to discuss the results and the proposed strategies  with city planners and architects to get their view on the topic as well.

Author Response

Reply to review #3

Main remarks:
1. Please write PET explicitly as "Physiologal Equivalent Temperature", this is not been done throughout the whole document (and there might be readers who are not familiar with this).
Ad. 1. We developed the concept of PET in the abstract and at the beginning of the main text in section "1.1. The urban heat island effect – causes, effects, and countermeasure".

2. In general, it would be good to have a table of abbreviations at the end of the paper, as there are many.
Ad. 2. Following the recommendations of the journal, all abbreviations firstly used in the abstract, tables, figures, and the main text have been fully explained.

3. For further research: it would be good to discuss the results and the proposed strategies with city planners and architects to get their view on the topic as well.
Ad. 3. We think it's a good idea. In the last paragraph of section "5.3. Future research direction", we have included our manifesto related to it. We will undertake activities to promote the discussion on the above subject in the academic, local and professional environment to obtain valuable opinions and consensus. We will also conduct educational and popularization activities based on the results of this study.